# A Systematic Review of Dynamic, Kinematic, and Muscle Activity during Gymnastic Still Rings Elements

**DOI:** 10.3390/sports11030050

**Published:** 2023-02-22

**Authors:** Roman Malíř, Jan Chrudimský, Michal Šteffl, Petr Stastny

**Affiliations:** Faculty of Physical Education and Sport, Charles University, 162 52 Prague, Czech Republic

**Keywords:** men’s artistic gymnastics, biomechanics, exercise technique, training load, strength

## Abstract

Still rings are a unique gymnastics apparatus allowing for a combination of dynamic and static elements in a specific technique. This review aimed to compile the dynamic, kinematic, and EMG characteristics of swing, dismount, handstand, strength, and hold elements on still rings. This systematic review was conducted in concordance with PRISMA in PubMed, EBSCOhost, Scopus, and Web of Science databases. In total, 37 studies were included, describing the strength and hold elements, the kip and swing elements, swing through or to handstand, and dismounts. The current evidence suggests that the execution of gymnastics elements on still rings and training drills requires a high training load. Specific preconditioning exercises could be used to train for the Swallow, iron cross, and support scale. Negative impacts of load during hold elements can be reduced by special support devices such as the Herdos or support belts. Another aspect is improving strength prerequisites by exercises such as bench presses, barbell exercises, and support belts, where the main focus should be on muscular coordination similar to the other considerable elements. Electromyography is an appropriate tool for the investigation of muscular coordination and force platforms for assessing a sufficient strength level to successfully perform elements on still rings.

## 1. Introduction

Compared with other disciplines of all-around men’s artistic gymnastics, rings are still a unique discipline, where the gymnasts are required to use their upper limbs to support their body mass [1] during the whole routine. The absence of a solid base due to movable rings connected to steel construction affects typical movements, training methods, performance assessment by judges, and injury risk. According to The Code of the points of Men’s artistic gymnastics (MAG CoP) [2], elements on still rings are officially divided into Element Groups, which are: kip and swing elements, and swing through or to handstand (hold 2 s); strength elements and hold elements (2 s); swing-to-strength hold elements; and dismounts, where this classification is critical for routine composition and its assessment of difficulty by judges. The performance in men’s artistic gymnastics is assessed as a sum of the difficulty score and execution score, which then make up the final score [3,4]. For each apparatus of men’s all-around gymnastics, the unique requirements for the composition of the routine, its difficulty, and assessment of technique errors are determined. The specific requirements and evaluation of technical errors in still ring apparatus result from discipline specification (moving steel cables) such as the wrong position of rings (hands turned in on L-sit) or excessive movements in static elements such as handstand after forward swing [5]. The gymnast has to involve at least one element from each element group in their routine. In particular, the strength elements and swing elements are critical for routine composition in order to gain a higher final score. On the other hand, perfect execution is required because all deviations from an expected way of execution are penalized based on MAG CoP [2]. For the inclusion of specific elements into the routine, the gymnasts should perform these elements at least to grade “3” or “4” according to an assessment of the quality value scale [4].

Exercising on still rings is included at an early stage of sports participation, training, and competition. The training on the still rings starts with more effortless strength static elements and swing movements at long hangs [4,6] and continues with the training of elements with a higher point of difficulty, which increases demands on technical preparation and physical prerequisites such as muscle strength, which have to be developed. High demands increase the weight-bearing on the upper extremities, mainly on shoulder joints. An adequately mastered technique requires a well-conducted training process based on quality diagnostic methods for evaluating the mechanical loading on the gymnast’s body. One way to evaluate loading on the gymnast’s body during the execution of exercises on still rings is by measuring external forces acting on the gymnast’s body [7]. In this way, it is easier to avoid unnecessary injuries that can endanger a gymnast’s health or their entire sports career. Within the strength and hold elements category, there are relative and maximal strength requirements for gymnasts, primarily in isometric and eccentric muscle contractions [8,9,10,11]. The relative strength of the gymnast could play an important role in Swallow, for instance [12]. On the other hand, Schärer [10] claimed that training under maximal muscle tension for a very long time improves the execution of Swallow, iron cross, and support scale; thus, the maximal and endurance strength could be needed [10].

Due to the high complexity, several training aids and methods serve as useful tools making the training process easier and safer. Training methods, such as strengthening exercises aimed at strength and coordination development of selected muscles used in specific elements, are preferably considered. For instance, Bernasconi [1] compared shoulder muscle activity during two exercises with barbells and during a support scale on still rings, with the result that barbell exercises reduce the participation of the serratus anterior in stabilizing the scapula [1]. The biomechanical approach can be the critical point of view for distinguishing the correct way of different gymnastics elements, for making training strategies or their valuation.

Several studies aimed to strength and power development had already been published; however, it is not clear what preconditioning exercises and types of muscle contraction are preferred for selected strength elements and hold elements with maximal strength, relative strength or strength endurance. Another potential disagreement is in external loading evaluation during backward and forward swings, where the external loading can take on different values.

Current studies provide a disunited description of gymnastic rings biomechanical characteristics and their uncertain movement pattern relations. Therefore, the aim of this review was to compile the dynamic, kinematic, and EMG characteristics of swing, dismount, handstand, strength, and hold elements on steady rings in any cross-sectional or intervention studies comparing any biomechanical measures on male gymnasts. This review should be able to distinguish an appropriate training method, performance evaluation, and correct technique evaluation based on MAG CoP [2]. 

## 2. Materials and Methods

This systematic review summarises the current scientific literature focused on gymnastics still rings and was conducted in concordance with the recommendations of PRISMA [13] (Figure 1) using the review protocol (available in Appendix A).

### 2.1. Search Strategy

Two researchers from author’s membership (RM and JCh) independently performed the literature search in the four databases on the 6th of February 2023: PubMed, EBSCO, Scopus, and Web of Science. For the search strategy, we used the same stream in all the databases (Table 1).

### 2.2. Eligibility Criteria

Only peer-reviewed papers were included, exclusively available in full and in the English language, reporting still rings’ exercise among a population of male artistic gymnasts of any age and any level. Due to the relatively low count of studies found, we included studies regardless of the date of publication. After removing duplicates, we also removed studies which did not relate to men’s artistic gymnastics and rings (678 removed); subsequently, we removed studies with no direct relationship to still rings exercises (4).

### 2.3. Data Extraction and Synthesis

Results from the four databases were put into a references manager, and then all the duplicate records were discarded. Two reviewers using the Excel^®^ spreadsheet (Microsoft Corp., Redmond, WA, USA) extracted data independently. Extracted information covered the country of origin, study design, study population (number of participants), main tools, focuses, and main findings. Both narrative and quantitative syntheses of findings from the included studies, structured around the types of exercise are provided in this review. For all studies that were subsequently selected, we performed risk of bias evaluation by JBI Critical Appraisal Checklist for Analytical Cross Sectional Studies.

### 2.4. Supplementary Online Material

The risk of bias of each study included in this systematic review, PRISMA checklist, protocol of the systematic review, and full searching formula (including .csv of PubMed and Scopus) are available in Appendix A.

## 3. Results

Out of all 721 papers found in the four databases, including the 3 papers from the references lists checking, and after assessing eligibility, we included 37 studies in the qualitative analysis. Overall, these studies included 263 elite gymnasts and 11 national coaches as participants. Studies came from 16 countries: 7 studies were from the United Kingdom (UK), 3 were from France, and United States (US), 5 from Switzerland, 2 were from Bulgaria, Poland, Canada, China, Brazil, and Spain, and one study each was from Slovenia, Argentina, Iraq, Germany, Japan, and Egypt. Dividing the studies according to the methods revealed that most were observation studies, and out of them, 9 were case studies. Only 2 studies were experimental, and 3 studies were quasi-experimental. Out of the main methods, the kinematic analysis was the most frequent (16×), followed by EMG (7×), force platform and force system measurements (5×), as well as muscle strength tests (4×). The performance level of elements on still rings was used as a measure only once. Biomechanical characteristics of selected elements were the most frequent focus (10×), followed by the comparison of different training conditions (5×), and the evaluation of the force platform and force system measurements as a training aid (7×). The verification of new training aids and methods was the main topic of the experimental studies. The risk of bias for each study included in this systematic review is listed in the Appendix A.

The strength elements and hold elements were included in 22 studies, where strength or gymnast net force (measured predominantly on force plates) were identified as differentiation factors between performers and non-performers of L-Sit cross [8], Maltese cross [14], Iron cross [15], and Swallow [9,16], Azarian [17], and Swallow-holding duration [12].

The elements which require high muscular coordination is possible to evaluate by EMG, specifically in Azarian cross [18], support scale [1,19], inverted cross [20], and iron cross [21,22]. The basic description of all the included studies is presented in Table 2.

Considerable external loading is present during swing elements as reaction forces are produced on the gymnast’s body, where the reaction forces are greater during back swing than forward swing [30]. On the other hand, the pulling forces produced by gymnasts during the backward and forward long swing are almost similar [31,32]. The external loading could be reduced by horizontal displacement of the gymnast’s centre of mass during swings, can reduce centrifugal interactive force and to minimize the mechanical load on the shoulder joints [33]. This is most evident when the gymnast is commencing the backward giant circle when the handstand swing reaches the bottom of its swing arc [34].

Asymmetrical arm movements contributed to the subsequent removal of tilt during twisting techniques used in dismounts [35]. A very high force and momentum is needed to perform triple salto backward from still rings with extreme grip strength during the preparation phase. A comparatively fast rotation is required during the flight phase with extreme strength and fast work of the trunk and leg flexors, as well as strength of the leg extensors [36]. The landing after the backward giant swing tuck of 2-circle back flip and 360° turn is characterized by increasing of the velocity of gravity centre with a swing down from the handstand with subsequent leg throw up forwards while the whole body passes through the vertical plane. Both hands left the rings immediately at the moment the rings resumed from the handstand [37]. A handstand on still rings is specific to different muscle activity such as the pectoralis major, latissimus dorsi, biceps brachii, descendent part of trapezius and the deltoid muscle due to instability of the rings [38], where special tools such as “rocker” or “bowls” might be used to train the handstand [26,27]. Two studies without the orientation to the specific elements on still rings were described as others. Brewin and Kerwin indirectly evaluated cable tension during specific static (1/2 lever) and dynamic (basic swing, inlocates and dislocates, backward and forward longswing) elements on still rings [7]. Kochanowicz et al. (2019) compared the EMG of specific muscles during handstand on three different apparatus (i.e., floor, parallel bars, and still rings) [38] (Table 3).

Four studies investigated dismounts on the still rings’ apparatus [35,36,37,40]. Some of the observed dismount’s elements were the triple salto backwards, full twisting double somersault, 2-circle back flip and a 360° twist, and double back straight somersault and double back straight somersault with a full twist. All four included studies were biomechanically pointed but with different aims. The basic description of dismounts is highlighted in Table 4.

We ultimately classified four studies as others. The first study was interested in the kinematic analysis of specific elements on still rings [7]. However, the study also sought to complete a kinematic analysis similar to previously included studies, this study also considered the combination of static and dynamic elements. The second study identified the muscle activity during a handstand on various apparatus including still rings [38]. The basic description of these two studies is highlighted in Table 5. The last two studies referred to describing the training plan of one Olympic gymnast [41], and descriptions of trends and guidelines across key exercises during the training perspective of still rings [42].

## 4. Discussion

Still rings are a unique apparatus due to requirements of advanced strength abilities in order to execute a constrained number of elements chosen from the element groups performed during the routine, as we listed above. The studies included in this systematic review provided information about various research approaches and outcomes focused on performance on still rings. A common feature of relevant studies is the performance level of participants, mostly elite gymnasts. An important issue was the gymnastics terminology that was not uniform across the reviewed studies. It seems that in different states and regions, various names for the same gymnastics elements are used (i.e., Maltese cross vs. iron cross or cross support). The comparison of research outcomes is complicated by utilizing uniform measurement units and their transformation (e.g., kN, kg, multiple of body weight).

### 4.1. Strength, and Hold Elements on Still Rings

Among the studies categorized as force and balance elements or strength abilities, seven focused on the Swallow, five iron cross, and three on the support scale. All of these elements are commonly used during the still rings routine. The substance of the Swallow element performance is in keeping the whole body in a horizontal plane at the height of the rings with straight arms for two seconds [4,43]. Iron cross performance is described as maintaining 90° shoulder abduction in the frontal plane with both straight upper limbs for at least two seconds [4]. The support scale is a typical strength element performed at the straddle legs position or with legs together by all top-level gymnasts. Including these elements is required at both age categories (Junior, Senior); the rules limit their number and order in the routine. Therefore, strength training and the training of these skills are a common part of young gymnasts’ training.

Within the evaluation of still rings routines during competitions, reliable measurement systems can help to minimize errors made by judges during gymnastic routine evaluation, especially for determining the holding time (2 s minimum) of strength and hold elements [29]. On the other hand, it also depends on the point of view [44,45] of the judges, who evaluate not only the holding time, but also, for example, deviations from the angles within the posture of the individual elements (90° shoulder abduction in the iron cross), which are determined by rules [45].

#### Training Methods, Preconditioning Exercises and Evaluation in Strength Elements and Hold Elements

In connection with the above, it seems that the training methods, utilization of precondition exercises, and different methods of gymnasts´ strength evaluation are the most important information for coaches.

Results of reviewed studies showed the suitability of using drills with spotting, counterweight, dumbbells, and other devices to increase the strength training effect [1,8,9,10,14,16,18,20,21,23,24]. For example, the counterweight exercise could be used to strengthen the muscles that stabilize the scapula (serratus anterior and trapezius), given proper muscle coordination during execution concerned element. The barbell exercise could provide an interesting exercise to prepare for muscle coordination of the shoulder flexors. The dumbbells exercise may be valuable in the initial preparation of rotator cuff muscles for the still rings because the load could be adapted to each shoulder [1]. It seems that some training tools or devices help with the exercises themselves on still rings, for example, “herodos”, which shortens the lever and, therefore, reduces stress on the elbows and shoulders of gymnasts [15,18,21]. The solution for specific muscle activity coordination during strength elements training seems to be a utilization of spotting by another person (trainer), such as during an inverted cross from below for stimulating the element [20]. Preconditioning exercises are a common and useful practice in gymnastics strength training. Hübner and Schärer [9] found a significant correlation between Swallow and preconditioning exercises, Swallow supine position and bench press. Additionally, specific training methods using specific muscle contractions could be considered within short training exposure. Eccentric exercises are often used to improve maximum strength, where the eccentric isokinetic training with additional load during training could be appropriate method to effectively increase maximum strength and strength endurance in short term training [11].

Results of reviewed studies showed the benefits of strength testing assess athletes’ strength abilities and distinguishing athletes who had sufficient strength status for considered high-assessed skill on still rings [14,15,25]. In this context, it has to be mentioned that coaches should constantly search to improve gymnasts’ strength by using readiness prediction and utilization of optimal strength methods. For this purpose, the force platforms could be appropriate devices. As some authors show [8,14,15,16] force platforms could be useful for holding elements (e.g., iron cross, Maltese cross, L-sit cross, Swallow, and others), where is necessary to compare gymnasts’ level of strength, and estimating their level of readiness to perform selected elements. Fujihara (2023) used specific force analysis system to measure the force generated by gymnast on rings during Azarian element. The system works on the principle of sensing the gymnast’s weight via cable tension and at the same time is able to calculate the percentage value of the weight being spotted by the coach. Unlike force plates, this system can be implemented directly on the rings [17]. Compared with the force plates and force analysis systems such as assessing tools for athletes’ strength abilities, EMG is also used in still rings exercises to detect muscle activity during selected elements/movements [1,18,20,21,22]. It seems that knowledge about muscle activity, and the co-contraction of activated muscles during performing different still ring elements could be useful for optimizing training methods not only for strength and holding elements. However, the positive effect of different training devices results from strength elements execution and the synergy of muscle groups, as shown, e.g., utilization of a special belt for Azarian seems to be suitable rather than “herdos”, whereas the “herdos” did not stimulate the same muscle coordination in the shoulder joint compared with the belt on still rings [18]. 

Preconditioning exercises are common praxes, but their utilization and level of specificity are discussed. As the results show, EMG could help to find variations in muscle activity during different types of preconditioning exercises, e.g., pectoralis major participated less in shoulder flexion during the counterweight exercise, whereas the deltoideus was more activated during dumbbells exercise, the barbell exercises reduced the activity of serratus anterior [1]. Göpfer et al. (2022) used EMG for description of the wavelet-transformed changes muscle intensity pattern and frequency spectra of eight upper body muscles during Swallow and support scale execution. The observed changes corresponded with acute muscular fatigue during both elements [19]. From this point of view, EMG could also be a suitable tool for testing changes in muscles patterns in other static elements on still rings.

However, strength and hold elements are only one part of the competition routine. The different elements (strength and swing elements) evoke repetitive concentric and eccentric muscle contractions and levels of their activation. This reflects the specific fatigue and recovery rates of different muscle groups in relation to fibre composition and metabolic demands [28].

### 4.2. Swings on Still Rings

Swing elements together with Kip form one group of exercises according to CoP FIG. Swing elements could be performed in different ways of execution. Senior and Junior gymnasts have to perform at least one swing element to the handstand with 2 s. hold in their routine and must be inside ten counting elements for seniors or eight for juniors [2]. The lonswings are a common part of young gymnasts’ training and coaches recognise them as crucial skills for the acquisition of other skills. Alasim et al. [46] claimed that pulling forces in the right and up direction require higher activation of shoulder muscles (supraspinatus and infraspinatus); thus the pulling forces could be detected during longswings. The acting forces during pulling phase are comparatively high (6.5 BW) [30], and can be the cause of injuries. Serafin et al. [30] concluded that the influence of reaction forces was significantly greater during back swings than during forward swings, and supported these findings by higher susceptibility of the motor system to forward bending rather than backward bending during swings. On the other hand, several study results showed the possibility of optimizing the gymnast’s technique which could contribute to reducing mechanical loading influencing the gymnast’s body. In particular, the horizontal displacement of the gymnast’s centre of mass during swings can contribute to minimizing the mechanical load on the shoulder joints [33]. On the other hand, we do not find a general agreement among the authors about the amount of the acting force on gymnasts’ body parts (mainly shoulders and hips) according to the direction of movement or the body position during exercise [31,32]. With regard above and other results, the main role for energy generating have hips and shoulders muscles. The hip flexors play an important role in excessive hyperextension prevention during downward swing phase and the shoulder flexors and extensors are primary sources of energy generation during whole movement [39].

The long swings often begin and end in a static position. Most frequently, the handstand is the initial and required final position for the giant circle. For that reason, the handstand was assumed as a critical skill for giant circle performance [27,38]. Mastering of the handstand at the competition level requires a long-term and systematic training process, and also the quality of execution is important. Gymnasts have to reduce actively the amplitude of oscillation rings during exercise to hold a static body position [4]. According to Sprigings et al. [34], the gymnast should eliminate any swing in the final or next handstand if an initial swing amplitude is in the range of 3–6°. Alongside this, the body configuration changes should be timed within 15 ms, whereas a delay of 30 ms would result in a considerable residual swing. Therefore, maintaining the initial and final handstand in a static position without apparent swings when performing backward giant circle is necessary [5].

The large number of elements performed during the training process may cause undesirable overload. A problem arises, especially when elements are performed with lower quality of technique or poor execution [33]. Improperly performed exercise increases the negative impact of training, which may lead to injuries [47,48,49]. Hart et al. [6] reported high injury rates in gymnastics sports and shoulder joints’ injuries were often associated with still rings [6,50]. Moreover, injuries of the upper extremities and especially the shoulders are characterised as injuries connected to fatigue or wrongly performed techniques [6,28,50,51,52]. Therefore, training according to current scientific knowledge (using all available training equipment) may help to keep gymnasts safe from injury. Beyranvand et al. [49] concluded that having rounded shoulders could significantly affect further stability of the upper limbs and increase injury risk. On this basis, we should also pay attention to the stability of the shoulder joint. The elasticity of the still rings’ apparatus, especially of the cables, may minimise mechanical load at the shoulder joints [33], which are often injured in men’s artistic gymnastics on the still rings’ apparatus [4,50]. Brewin et al. [33] showed that the flexibility of the gymnast and apparatus contributed to minimising peak shoulder forces. Despite the apparent considerable mechanical load in pulling forces acting on the gymnast’s body, there were just two studies that mention injury risk.

### 4.3. Dismount on Still Rings

Dismount is an integral part of all routines performed on still rings. The execution and difficulty contribute to the final score. In our review, only three studies investigated dismounts on the still rings’ apparatus [35,36,37,40]. All four included studies were biomechanically pointed but with different aim. Some observed dismount elements include triple salto backwards, full twisting double somersault, double back straight somersault with and without full twist, and 2-circle back flip and a 360° twist. Although these dismounts were very difficult elements even for elite gymnasts, coaches of elite gymnasts can profit from the results of their kinematic analyses. They could use the obtained results for training methods development, mainly if they are analysed elements with a high level of difficulty performed by top-level gymnasts. For instance, the magnitude forces at landing can range from 3.9 to 14.4 times a gymnast’s body weight, and the force magnitude is related to the skill difficulty [53]. That is a relatively considerable load, which the gymnast’s body has to resist, mainly when the loading acts repetitively from one attempt to another. The triple salto backward study was interested in the biomechanical characteristics of the element with specific strength and speed demands on the gymnast’s body during execution of the element [36]. Similarly sufficient speed of legs (ankles) is essential during double back straight somersault and double back straight somersault with full twist [40]. Although the study of Yeadon was aimed to biomechanical aspects of selected element too, the main findings corresponded to movements of arms during twist initiation in dismount. The author claimed that arms were predominantly in asymmetrical positions during twisting and consider arm movements as an important contributor to initiate twists during dismounts from still rings. The key points for coaching the removal of tilt are to pike prior to landing and to abduct the left arm if the twist is to the left [35]. The arms’ asymmetry is also in concordance with Kolimechkov et al. (2021), who claimed that during twisting within double straight somersault one of two gymnasts used slight asymmetrical actions of arms. Zou et al. aimed their research to landing after backward giant swing tuck of 2-circle back flip and 360° turns, especially for the improvement of dismount’s technique and theoretical advice for dismount. The author described speed differences during the swing phase where the greatest speed of the centre of gravity was crossing the vertical plane on the rings. The whole body should be in a straight position during the moment when the hands leave the rings. This effect speeds up the level of rotation. Further importance is assigned to the trunk angle during landing where too much forward lean of the trunk can lead to more forward step [37].

### 4.4. Handstand on Still Rings

It is clear that the handstand is an important element for many other gymnastics skills performed not only on the still rings. However, handstands on still rings are a bit more difficult than on other apparatus. The comparison of performing a handstand under different conditions (three apparatus: parallel bars, floor, and still rings) revealed higher EMG activity and, therefore, greater difficulty of handstand on still rings compared with the other apparatus (probably due to unstable still ring construction) [38]. Similar to strength training, the issue of usability, effectiveness, and specificity of various training devices is solved. Examples can be seen in the studies by Yeadon et al. [27] and Khargan et al. [26], who solved problems with still rings instability, especially during training. The authors in both studies designed, constructed, and assessed training aids that help to eliminate the still rings’ inherent swinging. The results of those two studies show that these aids simplify the training process and allow learning handstands on still rings more effectively. The explanation could be that special exercises with assistance training tools or devices have a great role in improving motor and skilful abilities [26,27], and their strong similarity (specificity) to final execution.

As part of sports preparation, careful periodization and planning of training is needed to achieve the best sports performance [54]. The issue of periodization of training is also crucial in gymnastic especially in still rings exercises, specific testing of strength level and subsequent detailed planning of training period can optimally increase the performance [41]. Another aspect is in the compliance of guidelines already at an early age of training. It is necessary to take into account some elements that are key to the evaluation of sports performance on the rings, but are also important for the future development of the gymnast. Yanev (2021) claimed that junior gymnasts should perform specific elements such as Jonasson and Yamawaki and swing elements to handstand within an increase in the D score above 4.5.

## 5. Conclusions

In conclusion, a relatively high interest in different performance aspects on still rings was found in this study with relatively high diversification across the included studies. Comparatively to studies aimed on different pieces of apparatus (i.e., high bar, floor, parallel bars), there are a lack of studies aimed at injury risk prevention. Only 2 studies out of the total of 37 pointed out a possible negative influence of the mechanical load on the risk of injuries, which is most likely associated with incorrect technique. Therefore, future research should be focused on injury issues related to exercises on still rings.

### Practical Applications

The findings of this systematic review have several practical applications for gymnastics coaches. Coaches should pay attention to specific strengthening methods and their combination, including using specific training tools or devices and spotting for developing proper and improving muscular strength for selected strength and balance elements. Those aspects could be essential for quick learning of proper techniques for the strength elements and encourage the distinctive strength demands of the included active muscles. For future development of training methods and gymnasts’ performance, progress is important to integrate modern research methods for the evaluation of training outcomes as part of coaches’ common practice.

In connection with swing elements on the still rings, coaches should pay attention to reducing the negative effects of mechanical loading exposure on the gymnasts’ body, especially to the shoulder joints, by directing them to the proper technique. The relation to higher injury risk is particularly evident by the evidence of the negative impact of repeated increased external forces on the gymnast’s body during the training of swing skills on the rings due to improper technique.

## Figures and Tables

**Figure 1 sports-11-00050-f001:**
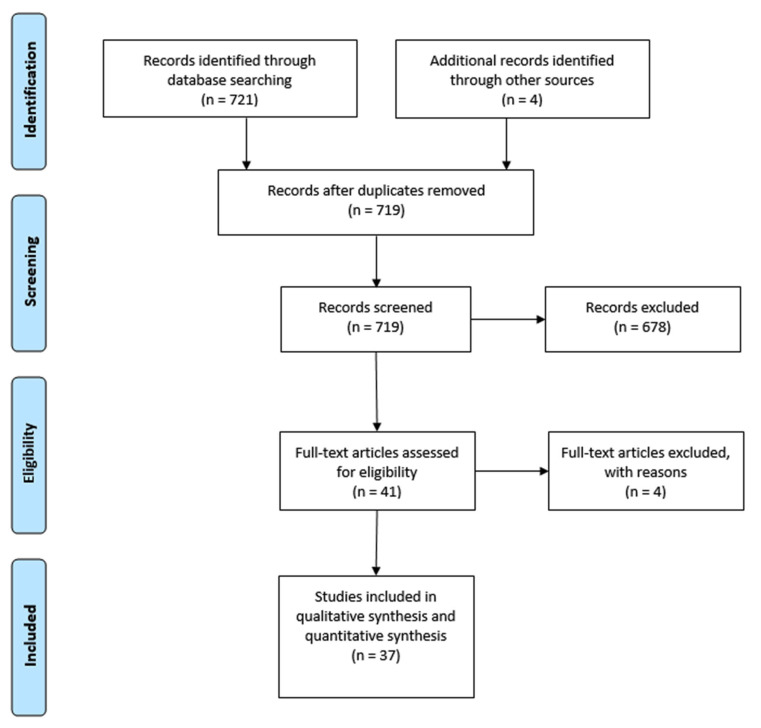
Flowchart illustrating phases of the search and study selection.

**Table 1 sports-11-00050-t001:** Search strategy.

Database	Key	Number
Web of Science	TOPIC: (gymnast *) AND TOPIC: (ring *)	96
Scopus	(TITLE-ABS-KEY (gymnast *) AND ALL (ring *))	408
PubMed	Search: “gymnast *” [All Fields] AND “ring *” [All Fields]	61
EBSCOhost	(TI gymnast *) AND (ring *)	156

**Table 2 sports-11-00050-t002:** The basic description of strength elements and hold elements.

Strength Elements and Hold Elements Study				
Study	Element	Methods and Participants	Main Tools	Main Focus and Conclusion
Bango et al., 2017, Spain, Observational study [8]	L-Sit cross	20 elite male gymnasts (age: 20.15 ± 3.29 years; weight: 68.53 ± 6.99 kg; height: 170.18 ± 6.38; experience: 14.1 ± 3.84 years)	Force platforms for evaluating readiness to L-Sit cross	Evaluation of practical application of the force platform as a training aid. The use of a single force platform can provide the gymnast and coaches sufficient information about how close a gymnast is to performing the L-sit Cross position, and can be used to monitor the training process in the case of less experienced gymnasts. The normalised maximum and mean isometric forces were extracted. Results showed large differences (*p* < 0.001, Cohen’s d = 1.6) between performer (P) and non-performer (NP) gymnasts of this element. P gymnasts produced a greater isometric strength level owing to their greater experience in training this element.
El-Awady, (2018), Egypt, Experimental study [23]	L-Sit cross	20 elite male gymnasts (10 experimental; age: 14.17 ± 0.7; weight: 47 ± 4.02; height: 147 ± 5.77), (10 control; age: 14.09 ± 0.8; weight: 49 ± 4.12; height: 145 ± 5.85)	Muscle strength tests and the performance level of LSC. TRX for improvements	Verification of new training methods and aids. Significant Difference between the experimental group and control group in Leg strength, Back strength, Core strength and performance level of L-sit cross for the experimental group. The functional suspension training with TRX for eight weeks could provide an improvement in the performance level of L-Sit cross.
Sands, (2006), USA, Observational study [14]	Maltese cross	16 elite gymnasts (age: 22.9 ± 2.3 years; height: 164.9 ± 6.0 cm; weight: 63.9 ± 7.6 kg)	Force platform	Evaluation of practical application of the force platform as a training aid. The strength testing approach described here was developed to ascertain the status and progress of elite male gymnasts. The initial results indicate that the method has sufficient fidelity to differentiate between athletes who can and cannot perform the Maltese cross.
Bernasconi, (2004), France, Observational study [21]	Iron cross	6 elite gymnasts (age: 23 ± 3 years; height: 167 ± 10 cm; weight: 65 ± 10 kg)	EMG	Comparison of different training conditions. Except for the M. teres major, the RMS decreases (*p* < 0.05) when using the Herdos. The SUM also decreases (*p* < 0.05) when using its device. The muscle parts indicate that the contribution of the M. latissimus dorsi decreases (*p* < 0.05) when using the Herdos. These results suggest that the Herdos modified the shoulder coordination. However, their usage does not seem to induce any functional adaptations of these muscles. The Herdos do not seem to provide a valid method to reproduce the same shoulder coordination as on the still rings.
Carrara, (2016), Brazil, Case study [22]	Iron cross	1 elite gymnast (age: 24 years; height: 156 cm; weight: 61.9 kg; experience: 17 years)	Kinematic analysis	Biomechanical characteristics of selected elements. Low variability values of shoulder angles and cable forces were verified and low values of asymmetry as well. Muscle activation varied according to muscle.
Dunlavy, (2007), USA, Observational study [15]	Iron cross	5 USA senior national team gymnasts (Performers of Iron cross; age: 23.8 ± 1.3 years; height: 159 ± 2 cm; weight: 66.6 ± 3.5 kg) 5 age group level gymnasts (Non performers; age: 14.0 ± 1.0 years; height: 160 ± 4 cm; weight: 55.3 ± 8.6 kg)	Force platforms	Evaluation of practical application of the force platform as a training aid. The mean and peak summed arm forces were able to statistically differentiate between athletes who could perform the cross from those who could not (*p* < 0.05). The force-time curves and small FPs showed sufficient fidelity to differentiate between the performer and non-performer groups. The force-time curves and small force platforms showed sufficient fidelity to differentiate between performer and non-performer groups. Force platforms may serve as useful adjuncts to athlete performance measurement.
Sands, (2006), USA, Case study [20]	Inverted cross	1 elite gymnast	EMG	The best drill for the inverted cross during performing is with a spot from below, and use a counterweight. The position should proceed in a closely simulated body position and with maximal to near-maximal intensity.
Schärer, (2016), Switzerland, Observational study [24]	Swallow, support scale and iron cross	10 elite gymnasts (age: 21.5 ± 2.5 years; height: 168.6 ± 4.5 cm; weight: 65.0 ± 5.0 kg; training time per week: more than 25 h)	Kinematic analysis and muscle strength tests. Method for predict maximum resistance	Comparison of different training conditions. A useful conversion table was established that predicts the maximum resistance at five and seven seconds holding time-based on the three seconds maximum resistance at each of the tested elements. The results showed a significant decrease in MR as holding time increased (*t*-test: *p* < 0.001). The standard error of estimate (SEE) and explained variance (R2) revealed that the prediction of MR at five seconds (SEE 0.52 kg to 1.03 kg, R2 0.92 to 1.00) was more accurate than at seven seconds holding time (SEE 0.95 kg to 2.08 kg, R2 0.88 to 0.98).
Hübner, (2015), Switzerland, Observational study [9]	Swallow, support scale and iron cross.	10 elite gymnasts (age: 21.5 ± 2.5 years; height: 168.6 ± 4.5 cm; weight: 65.0 ± 5.0 kg)	Muscle strength tests	Estimation of association between preconditioning exercises and performance of hold elements (correlation in strength between elements). A significant correlation was observed only between Swallow with the preconditioning exercises Swallow supine position (r: 0.71, *p*: 0.031) and Bench press (r: 0.71, *p*: 0.046); as well as between the Support Scale and Swallow supine position (r: 0.69, *p*: 0.039). Iron Cross correlated highest with the Cross belt (r: 0.66, *p*: 0.051) and bench press (r: 0.67, *p*: 0.069). Furthermore, it was observed that a minimal 1RM of 73.4% body weight is needed for the exercise Swallow supine position in order to complete a hold of the element Swallow on rings. For execution of the Support Scale element, a 1RM of 67.4% body weight for the exercise Swallow supine position is needed
Schärer, (2019), Switzerland, Quasi-experimental study [10]	Swallow, support scale	9 elite gymnasts (age: 21.47 ± 1.96 years; height: 169.84 ± 5.47 cm; weight: 69.4 ± 7.0 kg)	Kinematic analysis and muscle strength tests	Verification of a new training method. After four weeks of training, specific maximum strength increased significantly (Swallow: +4.1%; d = 0.85; *p* = 0.01; support scale: +3.6%; d = 2.47; *p* = 0.0002) and strength endurance tended to improve (Swallow: +104.8%; d = 0.60; *p* = 0.07; support scale: +26.8%; d = 0.27; *p* = 0.19). The high specificity but also the unfamiliar stimulus of slow eccentric movements with very long times under maximal muscle tension led to these improvements. To use this type of training periodically and during phases in which the technical training load is low.
Schärer, (2021), Switzerland, Quasi-experimental study [25]	Swallow, support scale, inverted cross	9 international and 10 national elite gymnasts (age: 22.03 ± 2.5 years; height: 169.38 ± 4.81 cm; weight: 64.99 ± 5.27 kg)	Kinematic analysis and muscle strength tests	Description of the relationship between a new conditioning strength test and a maximum strength test of static elements on rings in order to determine the minimal strength level (benchmarks) required to maintain these elements with one’s own body weight. High correlation coefficients were found between the conditioning maximum strength for Swallow/support scale (r: 0.65 to 0.92; *p* < 0.05) and inverted cross (r: 0.62 to 0.69; *p* > 0.05) and the maximum strength of the elements on rings. Strength benchmarks varied between 56.66% (inverted cross concentric) and 94.10% (Swallow eccentric) of body weight. Differences in biomechanical characteristics and technical requirements of strength elements on rings may (inter alia) explain the differences between correlations.
Bango, (2013), Spain, Observational study [16]	Swallow	8 elite gymnasts (Performers; *n* = 4; age: 24 ± 3.6; height: 165 ± 5 cm; weight: 630.41 ± 51.6 N), (Non-performers; *n* = 4; age: 17 ± 1.2; height: 171 ± 5 cm; weight: 662.25 ± 71.5 N)	Force platform	Evaluation of practical application of the force platform as a training aid. Results showed significant statistical differences between gymnasts that could perform the Swallow (P) from those that could not (NP) (*p* < 0.05). Performer gymnasts were characterized by a higher percentage of body weight descent and higher strength in relation to body mass (*p* < 0.05). The practical application of the force platform could be to provide coaches with information about how close the gymnast is to perform the Swallow.
Bernasconi, (2009), France, Observational study [1]	Swallow	6 elite gymnasts (age: 22 ± 3 years; height: 167 ± 6 cm; weight: 66 ± 8 kg; training time per week: more than 25 h)	EMG	Comparison of different training conditions. The counterweight exercise preserves the pectoralis major. The barbell exercise reduces participation of the serratus anterior (*p* < 0.05). The dumbbells exercise may be useful to prepare the rotator cuff muscles carefully for use.
Gorosito, (2013), Argentina, Observational study [12]	Swallow	14 elite gymnasts (age: 23 ± 4 years; height: 166.0 ± 5.0; weight: 67.8 ± 5.3 kg; sitting height: 87.8 ± 2.3 cm; wingspan: 176.0 ± 6.6 cm)	Swallow holding time, muscle strength tests	Estimation of the minimum relative strength required for a proper execution of the Swallow. A Spearman’s correlation test was used to compare the relative strength, height/sitting height and height/wingspan ratios versus the Swallow holding time of 14 senior Elite level male gymnasts from the Argentinean team. A significant correlation (*p* < 0.01) between the relative strength and the time in seconds that the Swallow was held by the athletes was found, proving that the execution of this element on rings is explained almost in a 90% by the gymnast’s relative strength. No correlation between the Swallow holding time and the height/sitting height and height/wingspan ratios were found.
Schärer, (2022), Switzerland, Quasi-experimental study [11]	Swallow and support scale	10 elite gymnasts (age 22.14 ± 2.99 years; height: 167.35 ± 4.07 cm; weight: 63.71 ± 4.04 kg; training time pre week: more than 25 h)	Kinematic analysis and muscle strength tests	Investigation of effect of three-week eccentric-isokinetic cluster training with a change of stimulus after three of six training sessions (eccentric-isokinetic with additional isoinertial load) on a computer-controlled training device on the improvement of the selected elements. Maximum strength and strength endurance were weakly determined. Significant increase was observed in maximum strength (Swallow: +8.72%; *p* < 0.001 and support scale: 8.32%; *p* < 0.0001) and strength endurance (Swallow: +122.36%; *p* = 0.02 and support scale: +93.30; *p* = 0.03). Three-week specific eccentric-isokinetic training with a change in stimulus after only three training sessions could be highly effective for improving the maximum strength and strength endurance.
Göpfer, (2022), Switzerland, Quasi-experimental study [19]	Swallow and support scale	8 international and national top-level gymnasts (age: 21.47 ± 1.96 years; height: 169.84 ± 5.47 cm; weight: 69.4 ± 7.0 kg)	EMG	Description of changes of the wavelet-transformed muscle intensity pattern and frequency spectra of eight upper body muscles during Swallow and support scale and subsequent effect of four-week eccentric-isokinetic intervention within different time intervals during the performance of Swallow and support scale was analysed. The EMG wavelet spectra presented changes corresponding to the performance gain with the eccentric training, and showed the frequency shift toward a predominant frequency due to acute muscular fatigue.
Bernasconi, (2006), France, Observational study [18]	Azarian	7 elite gymnasts (age: 21–26 years; height: 161–185 cm; weight: 56–83 kg)	EMG	Comparison of different training conditions. Results showed that muscles rhomboid, supraspinatus, deltoid (anterior, middle and posterior parts), biceps brachii and triceps brachii have significantly (*p* < 0.05) higher RMS when gymnasts are using the Belt than the Herdos. To conclude, if the Herdos and the Belt permit to reproduce the competitive movement, muscle activities are quite different between the two devices. The Herdos should, therefore, be used to more reduce the stress on the shoulder and elbow joints, whereas the Belt induces higher muscles activity and probably provides closer muscle synergisms to the rings.
Khargan, (2020), Iraq, Experimental study [26]	Handstand	8 young gymnasts (age: 132.00 ± 9.07 months; height: 139.37 ± 10.43 cm; weight: 33.75 ± 7.36 kg; training age: 52.50 ± 10.99 months)	Tests of the special motor abilities on the still rings	Verification of new training methods and aids. The special exercises that used assisting tools have an active role in the development of motor and skill abilities in the experimental group. Using the tools to assist in the process of improving the performance of some skills, which contributes to saving time and effort for the trainer and the player.
Yeadon, (2011), UK, Case study [27]	Handstand	1 elite gymnast and 11 national coaches	Kinematic analysis	Biomechanical designing of a new gymnastics training aid. As the training aid removed the inherent swinging of the rings, it simplified the handstand. It also simplified the balancing task by permitting the various degrees of freedom to be introduced separately and individually and successfully fulfilled all of the identified coaches’ requirements.
Irwin, (2002), UK, Observational study [28]	Rings routine	5 elite gymnasts (age: 21.0 ± 3.7 years; height: 173 ± 4 cm; weight: 70.4 ± 5.7 kg)	EMG	Description of muscle activation characteristics. The specific fatigue and recovery rates of different muscle groups, dictated by fibre composition and metabolic demands. Additionally, the specific requirements of the activity may have led to an increased contribution of specific muscle groups leading to further fatigue of these groups.
Lehmann, (2021), Germany, Observational study [29]	Rings routine	14 national team squad and non-squad gymnasts (age: 25.6 ± 2.9 years; height: 170.0 ± 6.2 cm; weight: 65.2 ± 4.9 kg)	Kinematic analysis	Investigation of two measuring systems for holding time evaluation. Two variants (dms10 and dms5) of dynamometric method were used as well as kinematic method (kms) based on a trained neural network were presented and examined with regard to their agreement with judges’ evaluations when measuring the hold time. The dms10 could be practicable and reliable method to assist judges in evaluating hold times but dms5 and kms were not suitable as means of judges’ support.
Fujihara, (2023), Japan, Observational study [17]	Azarian	2 university-student gymnasts (gymnast 1, weight: 56.2 kg; gymnast 2, weight: 60.1 kg)	Video and force recording	Real-time video and force analysis feedback system for learning strength and hold elements. The system is able to display the real-time video of performer on rings and can objectively measure amount of gymnasts’ support on the rings based on the weight of the gymnast. The system could be successful contributor to filling the gap between science and practice within in gymnastics.

**Table 3 sports-11-00050-t003:** The basic description of kip and swing elements and swing through or to handstand.

Kip and Swing Elements and Swing through or to Handstand Study				
Study	Element	Methods and Participants	Main Tools	Main Focus and Conclusion
Niu, (2000), China, Observational study [32]	Giant swing (longswings)	5 male junior elite gymnasts (age: 15.4 years; height: 154.8 cm; weight: 43.9 kg)	Kinematic analysis and telemetry EMG	Analysis of five giant swing phases. 1. With completion of the move as the body swings forward, the backward swing begins and the pulling force varies from 12.99 kg to 34.58 kg, lower than the body weight. This is the period when the gymnast will utilize potential energy 2. When the pulling force is greater than the body weight, both the centre of gravity of the body and the hip reach their maximum velocity. The former was between 3.08 m/s and 3.93 m/ s and the latter 3.23 m/s and 4.34 m/s. The lower back muscles such as gluteus maximus, the biceps femoris are fully contracted at this time. 3. The first peak value of the pulling force varies slightly between 182 kg and 207 kg, whereas the hip angle reduces to its minimum value of between 131° and 145°. The major muscles are fully stretched, and the giant swing begins. 4. At the second peak, the value of the pulling force was between 300 kg and 349 kg, the greatest among all phases. The time when the maximum components of the force are generated is between 4.50 ms and 13.00 ms after the vertical plane. This period presents a challenge for the performer to utilize potential kinetic energy. There were similar patterns in pulling force, shoulder angle, hip angle, hip velocity and ankle velocity when performing the movements of backward swing phase, dropped shoulder, giant-swing, and upward swing phase.
Sprigings, (1997), Canada, Case study [34]	Backward giant circle (backward longswing)	1 elite gymnast (height: 160 cm; weight: 58 kg)	Kinematic analysis	Biomechanical characteristics of selected elements. The optimal initiation of a backward giant circle is when gymnast’s swinging handstand has reached the bottom of its swing-arc, for a handstand with an original swing-amplitude of 10 degrees. An adequately timed backward giant circle can reduce this amplitude to a negligible 1.5 degrees of swing.
Sprigings, (2000), Canada, Case study [39]	Backward giant circle (backward longswing)	2 elite gymnasts	Video and force recording	Biomechanical characteristics of selected elements. The hip-joint flexors/extensors functioned as the primary source of energy generation to the system. From a swinging handstand, with an initial handstand swing amplitude of 16°, the gymnasts were able to arrive at the next handstand position with approximately 6–7.5° of residual swing, which was close to the optimal value of 4° predicted by computer simulation.
Yeadon, (2003), UK, Case study [5]	Backward longswing to handstand	1 elite gymnast	Kinematic analysis	Biomechanical characteristics of selected elements. For a final handstand with minimal residual swing, the changes in body configuration must be timed to within 15 ms, whereas a delay of 30 ms will result in a considerable residual swing. The lateral arm movements may provide the gymnast with more opportunities to make the task of performing the backward longswing easier and therefore contribute to a successful performance.
Brewin, (2000), UK, Case study [33]	Backward longswing to handstand	1 elite gymnast	Kinematic analysis	Evaluation of a gymnast’s technique and apparatus influence. During the evaluated longswing the peak combined force at the shoulders was 8.5 bodyweights. Modifications to the evaluated simulation of the longswing were used to determine the effect of the gymnast’s technique, his elasticity and that of the ring’s apparatus on peak net shoulder forces. Altering the gymnast’s technique, by fixing the gymnast in a straight body configuration throughout the swing, increased the peak shoulder force by 2.56 bodyweights. Removing lateral arm movements, which form part of the gymnast’s technique, also resulted in an increased peak shoulder force (0.73 bodyweights). Removing the elasticity of the apparatus and gymnast in turn resulted in smaller increases in peak shoulder force (0.62 and 0.53 bodyweights). When both aspects of the technique were altered, the increase in peak shoulder force was 2.5 times greater than when both components of elasticity were removed. The contribution of a gymnast’s technique is considerably greater than the contribution of the elasticity of the apparatus in minimising peak shoulder force.
Mills, (1998), UK, Case study [31]	Backward longswings, forward longswings and basic swings	1 elite gymnast	Kinematic analysis	Development of an indirect video-based method. The indirect video-based method was able to estimate cable tension to an accuracy of approximately 2 percent of the overall force range. This method is able to provide detailed information on the forces exerted on the rings during gymnastic movements performed in the competition.
Serafin, (2008), Poland, Case study [30]	Forward and backward swings	1 elite junior gymnast (age: 14; height: 161 cm; weight: 53.1 kg)	Cable reaction force and videotape recording	Biomechanical characteristics of selected elements. They amount to 5.5 BW for the forward swing and 6.5 BW for the backward swing movement. The maximum rate of change of the force for forward and backward swing is 42.6 BWs^−1^ and 67.4 BWs^−1^, respectively. These two variables differentiate the mechanical loading of the gymnast’s motor system between forward and backward swings. The reaction force produced by the gymnast was significantly greater during the execution of backward swings. The horizontal displacements of the gymnast’s centre of mass might be the factor responsible for the reduction in mechanical loading.

**Table 4 sports-11-00050-t004:** The basic description of dismounts.

Dismounts Study				
Study	Element	Methods and Participants	Main Tools	Main Focus and Conclusion
Čuk, (2010), Slovenia, Case study [36]	Triple salto backwards	1 elite gymnast (height: 169 cm; weight: 62.1 kg)	Kinematic analysis	Biomechanical characteristics of selected elements. Execution of the triple salto backwards requires extreme grip strength in the preparation phase and during the flight; extreme strength and fast work of the trunk and leg flexors; and extreme strength of the leg extensors. The triple salto backward is characterised by very high force (11.70 G) and momentum (4617 Nm) on the rings in the preparation phase, a very fast rotation around the x axis during the flight (860 °/s), a very small moment of inertia during the flight and it requires extreme grip strength in the preparation phase and during the flight (pulling knees as close as possible to the trunk), extreme strength and fast work of the trunk and leg flexors (receiving into and maintaining a tucked position), as well as extreme strength of the leg extensors (landing from 3.18 m).
Yeadon, (1994), UK, Observational study [35]	Full-twisting double somersault	6 elite gymnasts	Kinematic analysis	Biomechanical characteristics of selected elements. Symmetrical movements made substantial contributions to the removal of tilt, indicating that piking prior to landing automatically helped to reduce the angle of tilt. Arm asymmetries also made significant contributions toward the reduction in tilt.
Ningxiang, (2012), China, Case study [37]	2-circle back flip and a 360° turn	1 elite gymnast	Kinematic analysis	Biomechanical characteristics of selected elements. The velocity of the gravity centre increased as the gymnast’s handstand started to swing down. The velocity of the centre of gravity of his body moved up swiftly as his legs threw up forwards while the whole body passed through the vertical plane. Both hands left from the rings immediately at the point where the rising body resumed from the handstand. The landing angle was a bit larger because of a big step forward to make a stable landing; The landing skill needs to be improved.
Kolimechkov, (2021), UK, Bulgaria, Case study [40]	Double back straight somersault and double back straight somersault with full twist	2 elite gymnasts (gymnast 1, height: 169 cm; weight: 62 kg, gymnast 2, height: 163 cm; weight: 62 kg)	Kinematic analysis	Ankle speed of gymnast 1 and gymnast 2 during the execution phase were 11.11 m/s and 11.29 m/s, respectively. The angular velocity increased to 10.0 rad/s (gymnast 1) and 9.05 rad/s (gymnast 2). Gymnast 2 used small arm asymmetry during twisting technique with the beginning just before releasing the rings. Powerful pull combined with arching to piking beneath the rings and sufficient swing of the legs, are essential for successful execution of the dismount.

**Table 5 sports-11-00050-t005:** The basic description of studies assigned as “others”.

Study Considered as “Others”				
Study	Element	Methods and Participants	Main Tools	Main Focus and Conclusion
Brewin, (2003), UK, Case study [7]	Variety of static balances and dynamic swinging movements	2 elite gymnasts	Kinematic analysis	Evaluation of a gymnast’s technique and apparatus influence. The indirect video-based technique developed in this study accurately estimates combined cable tension throughout movements on rings. The indirect video-based technique developed in this study accurately estimates combined cable tension throughout movements on rings and may be considered for studies where a remote measurement is required.
Kochanowicz, (2019), Poland, Observational study [38]	Handstands performed on 3 apparatus (floor, rings, and parallel bars)	10 adult gymnasts (age: 25 ± 3.94 years; height: 172. 3 ± 4.3 cm; weight: 71.5 ± 2.99 kg; training work per week: 24 h; training experience: 17.8 ± 2.8 years) 15 young gymnasts (age: 13.9 ± 0.7 years; height: 154. 9 ± 9.8 cm; weight: 45.2 ± 7.7 kg; training work per week: 22 h; training experience: 7.7 ± 0.8 years)	EMG	Comparison of handstands performed on 3 apparatus. The different gymnastic apparatus led to specific muscle activation. This activation predominantly depended on hand support conditions, which alternated the primary wrist strategy of the handstand balance control, and in consequence, the activation of other muscles controlling balance.
Goto, (2022), Brazil, Case study [41]	Description of training plan and strategies by within training sheets, strength tests, macrocycles and microcycle	1 elite (Olympic medallist) gymnast (age: 22 years; practice: 13 years; training work per week: 25–30 h)	Observation/description	The periodization of training and careful planning led to increase the complexity of the technical elements of routine of the gymnast. The main factors influencing the achievement of the Olympic result were the training periodization in three stages and the competitive tactics in the preparatory evaluations
Yanev, (2021), Bulgaria, Observational study [42]	Description of trends and guidelines across exercises that are key to the development of gymnasts at an earlier age.	8 junior finalists in the ring final from the 1st Junior World Artistic Gymnastics Championships in Gyor, Hungary	Observation/description	The elements of higher frequency were from C group of difficulty, and the majority of the exercises were from EG I. All gymnast performed Jonasson and Yamawaki from EG I. Coaches and junior gymnasts should try to increase D score above 4538 by selecting swing and swing to handstand elements from EG I.

## Data Availability

Data will be made available upon request.

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
