# Peer review of "A Systematic Review of Dynamic, Kinematic, and Muscle Activity during Gymnastic Still Rings Elements"

_sports, 2023, doi:10.3390/sports11030050_

Round 1

Reviewer 1 Report

The authors presented a good revision of the literature about the gymnastics ring performance and training studies. This review has been well structured and presented interesting results and conclusions. However, several aspects must be accomplished, mainly related to the PRISMA requirements to do a systematic review:

INTRODUCTION

1.       Lines 81-83: The aim of the study has to refer explicitly to Participants, Interventions, Comparisons, Outcomes, and Study design (PICOS).

2.       Lines 84-85: These lines reflect the possible practical applications of the review but seem to go further from the aim of the review.

3.       Lines 85-89: Hypotheses are not related with the aim presented in the study. In addition, they are not used to discuss later in the manuscript.

METHODS

1.       Did the authors create a review protocol previously to conduct the systematic review? If that is the case, please, indicate the web address and the registration number.

2.       Figure 1: The authors did not conduct a meta-analysis. Please, in order to avoid misunderstandings, delete “(meta-analysis)” from the last box of the flowchart.

3.       Lines 96-100 and Table 1: Complete information of the search strategy presenting as a supplementary file the full electronic search strategy for at least one database (PubMed, for example).

4.       The authors did not include a “Risk of bias of individual studies” and it is also an important part of the systematic review to evaluate the findings robustness of the review. Please, include it and describe the method used.

RESULTS

1.       Present data on risk of bias of each study and, if it possible, across studies.

I also recommended to include the PRISMA checklist (https://www.prisma-statement.org/documents/PRISMA_2020_checklist.pdf?AspxAutoDetectCookieSupport=1) as a supplementary files

Here you have a recent Systematic Review published in the same journal (MDPI Sports) that can be used as an example:

Riquelme-Hernández, C.; Reyes-Barría, J.P.; Vargas, A.; Gonzalez-Robaina, Y.; Zapata-Lamana, R.; Toloza-Ramirez, D.; Parra-Rizo, M.A.; Cigarroa, I. Effects of the Practice of Movement Representation Techniques in People Undergoing Knee and Hip Arthroplasty: A Systematic Review. Sports 202210, 198. https://doi.org/10.3390/sports10120198

Reviewer 2 Report

PLEASE SEE ATTACHED
